# Highly Efficient Regeneration of *Bombax ceiba* via De Novo Organogenesis from Hypocotyl and Bud Explants

**DOI:** 10.3390/plants14132033

**Published:** 2025-07-02

**Authors:** Yamei Li, Qionghai Jiang, Lisha Cha, Fei Lin, Fenling Tang, Yong Kang, Guangsui Yang, Surong Huang, Yuhua Guo, Junmei Yin

**Affiliations:** 1Tropical Crops Genetic Resources Institute, Chinese Academy of Tropical Agricultural Sciences/National Key Laboratory for Tropical Crop Breeding/Key Laboratory of Crop Gene Resources and Germplasm Enhancement in Southern China, Ministry of Agriculture and Rural Affairs, Haikou 571101, China; lym6137@163.com (Y.L.);; 2School of Tropical Agriculture and Forestry, Hainan University, Haikou 570228, China; 3College of Tropical Crops, Yunnan Agricultural University, Pu’er 665099, China; 4Haikou Experiment Station, Chinese Academy of Tropical Agricultural Sciences, Haikou 571101, China; 5Sanya Research Institute, Chinese Academy of Tropical Agricultural Sciences, Sanya 571101, China

**Keywords:** *Bombax ceiba*, de novo organogenesis, hypocotyls, shoot buds, in vitro propagation

## Abstract

*Bombax ceiba* is an important medicinal and ornamental tree widely distributed in tropical and subtropical areas. However, its seeds lose viability rapidly after harvest, which has created hurdles in large-scale propagation. Here, we describe the development of a rapid and efficient de novo organogenesis system for *Bombax ceiba*, incorporating both indirect and direct regeneration pathways. The optimal basal medium used throughout the protocol was ½ MS supplemented with 30 g/L glucose, with all cultures maintained at 26–28 °C. For the indirect pathway, callus was induced from both ends of each hypocotyl on basal medium supplemented with 0.2 mg·L^−1^ 2,4-dichlorophenoxyacetic acid (2,4-D) and 0.5 mg·L^−1^ 6-Benzylaminopurine (6-BA) under dark conditions. The induced calluses were subsequently differentiated into adventitious shoots on basal media containing 0.5 mg·L^−1^ Indole-3-butyric acid (IBA), 0.15 mg·L^−1^ Kinetin (KIN), and 1 mg·L^−1^ 6-BA under a 16 h photoperiod, resulting in a callus induction rate of 140% and a differentiation rate of 51%. For the direct regeneration pathway, shoot buds cultured on medium with 0.5 mg·L^−1^ IBA and 1 mg·L^−1^ 6-BA achieved a 100% sprouting rate with a regeneration coefficient of approximately 3.2. The regenerated adventitious shoots rooted successfully on medium supplemented with 0.5 mg·L^−1^ Naphthylacetic acid (NAA) and were acclimatized under greenhouse conditions to produce viable plantlets. This regeneration system efficiently utilizes sterile seedling explants, is not limited by seasonal or environmental factors, and significantly improves the propagation efficiency of *Bombax ceiba*. These optimized micropropagation methods also provide a robust platform for future genetic transformation studies using hypocotyls and shoot buds as explants.

## 1. Introduction

*Bombax ceiba* L., commonly known as cotton tree, is widely distributed across tropical and subtropical regions worldwide [1]. In early spring (typically March in southern China), *Bombax ceiba* trees bloom profusely with striking red flowers before leaf emergence, creating spectacular landscape scenery. As a deciduous species, the tree sheds its leaves in the winter and can reach heights of up to 25–40 m [2]. The species also hold considerable ecological, medicinal, and economic value [3,4,5]. Its flowers are used to treat inflammatory diseases such as dysentery and enteritis [4], and the bark and root extracts are employed in traditional medicine to relieve rheumatism and menstrual disorders [5,6]. Industrially, its fruit fibers are used as fillers, and its seed oil has applications in lubricants and resins [7]. The hollow fiber, known as “soft gold of plants”, is valued in textiles and bio-composites [8], and the trunk is commonly used in sculpture, promoting seedling cultivation for commercial purposes [9].

Despite its broad application value, propagation protocols for *Bombax ceiba* remain limited. Traditional methods rely on seasonal seed availability, which is increasingly constrained by environmental degradation and habitat loss [10,11,12]. Moreover, the seeds exhibit short viability, inconsistent germination rates, and high susceptibility to pathogen infection and environmental stress, all of which further restrict its natural regeneration [13].

Plant tissue culture offers a solution by enabling the propagation of elite genotypes year-round, independent of environmental constraints [14,15]. It also facilitates somaclonal variation, mutation breeding, and genetic transformation, key tools in modern plant biotechnology. However, existing studies in *Bombax ceiba* tissue culture are limited in scope. Some have employed shoot tips from sterile seedlings and MS-based media with plant growth regulators (PGRs) such as 6-Benzylaminopurine (6-BA), Naphthylacetic acid (NAA), and Indole-3-butyric acid (IBA) for shoot proliferation [16,17,18], while others developed a somatic embryogenesis system using immature zygotic embryos [19]. Although these efforts represent progress, they rely on single regeneration pathways and lack a comprehensive, efficient system that utilizes multiple explants and supports high-frequency regeneration.

De novo organogenesis—both direct and indirect—is a widely applied method in plant biotechnology for shoot regeneration and is foundational to transformation protocols [20,21,22]. With the recent assembly of the *Bombax ceiba* genome and the identification of numerous functional genes [8,23,24], there is an urgent need for a robust regeneration system to support gene function studies and genetic improvement. Without such a system, efforts in genome editing, gene validation, and stress-resilience breeding are significantly hampered.

In this study, we present a novel regeneration protocol for *Bombax ceiba* that integrates both indirect and direct organogenesis using two explant types—hypocotyls and shoot buds. Using ½ MS supplemented with 30 g·L^−1^ glucose as the basal medium, we tested combinations of PGRs, including 2,4-dichlorophenoxyacetic acid (2,4-D), 6-BA, NAA, IBA, and Kinetin (KIN) across various stages: callus induction, shoot differentiation, adventitious shoot formation, rooting, and acclimatization. Our findings identify optimal medium compositions and culture conditions for each phase, enabling efficient and large-scale propagation of *Bombax ceiba*. This work not only contributes to conservation and commercial cultivation, but also provides a foundational platform for future genetic transformation and functional genomics in this valuable species.

## 2. Results

### 2.1. Optimal PGRs, Basal Medium, and Time for Callus Induction

Based on previous research, our study used MS as the base medium with added 2,4-D to induce callus formation from the hypocotyls. Our experimental results showed that increasing the 2,4-D concentration above 0.5 mg·L^−1^ 2,4-D can cause the explants to exhibit severe browning. However, MS combined with 0.1–0.5 mg·L^−1^ 2,4-D and 0.5 mg·L^−1^ KIN was able to partially induce callus, and the best induction was observed with medium that contained 0.1 mg·L^−1^ 2,4-D and 0.5 mg·L^−1^ KIN. This medium yielded an average callus induction rate of approximately 62.2% (Table 1). However, after 30 days of induction, the callus appeared watery and had a loose structure (Figure 1(Aa)). We also tested a combination of MS medium with 0.5 mg·L^−1^ 6-BA and 2 mg·L^−1^ NAA [25], but no significant callus formation was observed after 30 days. Additionally, the hypocotyls exhibited browning and root formation (see Figure 1(Ab)). Our results indicate that the combination of the plant growth regulator 2,4-D and the cytokinin 6-BA is more effective for callus induction. Further, when the 2,4-D concentration was maintained at 0.1 mg·L^−1^, the induction effect of 0.5 mg·L^−1^ 6-BA was superior to that of 0.2 mg·L^−1^, and the calluses that were formed were more enlarged and compact (Figure 1(Ac)).

Our previous research found that a basic medium and carbohydrate source were crucial for seedling growth [26]. Therefore, we also investigated the effects of two PGRs on callus induction and differentiation. When sucrose was used as the carbohydrate source in the culture medium, the explants were prone to browning. Although the difference in browning was not very pronounced, glucose was used as the primary carbohydrate source in subsequent experiments.

Compared with the full-strength MS basic media, ½ MS was more conducive to callus induction at both ends of the hypocotyls. As shown in Table 1, the callus induction rate increased significantly from 102% to 140% when using CIM5, which only differs from CIM4 in its use of ½ MS instead of MS as the basal medium. Taken together, our data show that ½ MS was the optimal basal medium for callus induction when combined with 30 g·L^−1^ glucose, 0.1 mg·L^−1^ 2,4-D, 0.5 mg·L^−1^ 6-BA, and 8 g·L^−1^ agar.

In addition, the induction period significantly affected callus status. In CIM5 medium, callus initiation from the hypocotyls began after 15 days of culture. The best callus growth was also achieved after 25 days of treatment. No significant proliferation was observed beyond this point, and gradual browning occurred after 30 days. Altogether, these results suggest that the optimal induction period for callus formation is between 25 and 30 days (Figure 1B).

### 2.2. The Effect of PGRs on Callus Differentiation

Next, we assessed the effects of IBA, KIN, and 6-BA on callus differentiation, with the results summarized in Table 2. Calluses were cultured under light conditions in different differentiation media for 30 days. However, not all calluses successfully differentiated into bud points; most underwent browning and eventually died. After 45 days, calluses cultured in DFM2 and DFM5 media induced adventitious shoots, and the DFM5 media caused significantly better differentiation compared with DFM2 (Table 2). On DFM5 medium, 46 out of 90 calluses, corresponding to a differentiation rate of 51.1%, differentiated into adventitious shoots, yielding a differentiation coefficient of 1.8. In comparison, only 4 out of 90 calluses on DFM2 medium differentiated, resulting in a differentiation rate of 4.4% and a coefficient of 1.3 (Table 3). As shown in Figure 2, calluses cultured on DFM5 for 45 days exhibited superior adventitious shoot differentiation compared with those on DFM2. These data indicate that both excessively high and low concentrations of the cytokinin 6-BA are unfavorable for callus differentiation, whereas 1 mg·L^−1^ 6-BA supplemented with a low concentration of auxin IBA promotes this process. However, it should be taken into consideration whether another cytokinin, KIN, may accelerate callus tissue browning.

### 2.3. The Effects of 6-BA and IBA on Direct Organogenesis from Bombax ceiba Buds

Based on our callus differentiation experiments, DMF5 exhibited superior differentiation efficiency and was therefore selected for adventitious shoot proliferation assays, with DMF10 used as a comparative medium (Table 4). Shoot buds derived from seedlings and tissue culture-induced adventitious shoots were used as explants to induce adventitious shoot proliferation via the direct organogenesis pathway on both DFM5 and DFM10 media. Both media showed strong proliferation capacity (Figure 3A,B), with sprouting rates reaching up to 100% and a proliferation coefficient of 3.2 (Table 4). Adventitious shoot proliferation was initiated approximately 20 days after inoculation and by day 40, the shoots had reached heights of 2–3 cm (Figure 3).

Notably, the combination of IBA and 6-BA in DFM10 produced a slightly better proliferation response than that observed in DFM5, although the difference was not statistically significant (Table 4). This trend may be associated with reduced browning at the basal region of the shoots (Figure 3). The improved performance in DFM10 could be attributed to its specific hormone composition, in which the concentration of the cytokinin 6-BA was twice that of the auxin IBA, and no KIN was present—conditions that may have favored the differentiation of explants into adventitious shoots.

Furthermore, shoot buds obtained from tissue-cultured plantlets were capable of undergoing another round of adventitious shoot proliferation after 30 days when cultured in the same proliferation medium. Ultimately, this process can be repeated until a sufficient number of plantlets are obtained, making the method highly suitable for large-scale propagation.

### 2.4. NAA Promotes Rooting of Bombax ceiba Adventitious Shoots

In order to understand which PGRs promoted the rooting of *Bombax ceiba* adventitious shoots, we tested NAA and KIN. Adventitious shoots were placed vertically into rooting media, R1 and R2. As shown in Table 5, both media induced root formation after 15 days of culture. After 30 days, the adventitious shoots that were inserted into R1 medium that contained 0.5 mg·L^−1^ NAA exhibited the best rooting induction. It showed a rooting rate of 96.7% and produced approximately three primary roots that were each 1–3 cm in length with abundant lateral roots. In contrast, the rooting rate in the R2 medium was approximately 64.4%, and produced only 1–3 shorter primary roots with fewer lateral roots. Compared with R2, shoots rooted in R1 medium were healthy and more elongated, with no browning at the shoot tips (Figure 4A,C). The primary adventitious roots were also thicker, longer, and had abundant lateral roots (Figure 4B,D). Ultimately, 0.5 mg·L^−1^ NAA was more favorable for adventitious shoot rooting.

### 2.5. Plantlets Growth in Soil

Finally, the rooted plantlets were thoroughly rinsed with tap water to remove residual agar and transplanted into pots containing a substrate mixture of nutrient soil, perlite, and vermiculite in a 3:1:1 ratio. After a 4-week acclimatization period in a growth chamber maintained at 25 °C, with a 16 h photoperiod and 60% relative humidity, the plantlets exhibited robust growth with fully expanded leaves, and the survival rate reached approximately 75% (Figure 5A). The acclimatized plantlets were transferred to soil; after four months, the plantlets developed into vigorous individuals (Figure 5B).

## 3. Discussion

In this study, we aim to develop an efficient and multi-pathway regeneration system for *Bombax ceiba* plantlet production. By using sterile seedling hypocotyls and shoot apical meristems as explants, we successfully established both indirect and direct organogenesis pathways. The primary goal was not only to rapidly and efficiently obtain *Bombax ceiba* tissue culture plantlets, but also to develop a standardized and reliable protocol that supports the large-scale propagation of elite *Bombax ceiba* germplasm resources. Importantly, this protocol also lays a theoretical foundation for future studies in the genetic transformation and molecular breeding of *Bombax ceiba*.

Currently, *Bombax ceiba* propagation mainly relies on seeds and cuttings. However, seed propagation suffers from low stability and is highly influenced by seasonal and geographical variations. Cutting propagation, on the other hand, is limited by branch availability and genotype-specific rooting potential, resulting in low efficiency. Tissue culture offers a promising alternative for large-scale propagation, providing a controlled environment, high multiplication efficiency, and improved antiviral immunity. Despite being influenced by explant type and genotype, tissue culture remains a valuable complement to traditional methods, particularly in enhancing plantlet quality and secondary metabolite production [27,28,29,30].

Previous studies on *Bombax ceiba* regeneration primarily used apical buds from sterile seedlings [18] and immature zygotic embryos [19], with most focusing on explants containing meristems [31,32,33,34]. In line with this, the buds used in our study were rich in apical meristem tissues. In addition, the vascular and cork cambia in the hypocotyls provided abundant lateral meristematic tissues composed of undifferentiated cells with sustained proliferative capacity [35]. Hypocotyls have been widely used as explants for regeneration in several major crops, including tomato (*Solanum lycopersicum*) [36], *Brassica carinata* [37], and cotton (*Gossypium*) [38], due to their high regenerative potential. This served as the main rationale for selecting hypocotyls as explants in our study. By leveraging sterile seedlings and optimized combinations of PGRs, we successfully induced callus formation from *Bombax ceiba* hypocotyls, thereby demonstrating somatic cell reprogramming in this species [39].

We achieved indirect organogenesis regeneration via the hypocotyl with a differentiation rate of 51% and a differentiation coefficient 1.8. In contrast, direct organogenesis from shoot buds achieved a 100% sprouting rate and an average proliferation coefficient of 3.22. Notably, the shortest regeneration cycle was completed in just 45 days. Furthermore, regenerated adventitious buds were viable as secondary explants, allowing cyclic regeneration. These results support the establishment of a reliable regeneration system and lay the groundwork for potential genetic transformation of *Bombax ceiba*, akin to that of cotton, using shoot apices or hypocotyls as target tissues [40,41].

The composition of the basal medium is a critical factor influencing growth and in vitro regeneration [42]. While MS medium has been widely used in previous *Bombax ceiba* tissue culture studies [16,17,18,19], our preliminary experiments revealed that full-strength MS often leads to browning and seedling mortality. Similar symptoms were observed during callus induction and shoot regeneration. In contrast, ½ MS significantly improved the callus induction efficiency and reduced browning. This improvement may be attributed to the lower concentrations of ammonium and nitrate in ½ MS, as excess nitrogen ions can induce oxidative stress, disrupt cellular homeostasis, and promote tissue necrosis [43]. Although we did not measure ion concentrations or osmotic potential in this study, our observations are consistent with similar findings in other species, where reduced salt strength alleviated stress-related browning [44,45]. We have acknowledged this limitation and emphasized the need for future physiological assays to verify this hypothesis.

In addition to inorganic salt concentration, carbohydrate metabolism also plays an essential role in plant regeneration. Sucrose is the most commonly used carbon source in tissue culture; however, our previous work demonstrated that replacing sucrose with glucose significantly enhanced germination and reduced root browning in *Bombax ceiba*. Therefore, glucose was used as the sole carbohydrate source throughout all stages of this study, including callus induction, shoot differentiation, and rooting. Beyond its role as an energy and carbon source, glucose can act as a signaling molecule that regulates the expression of genes involved in complex plant growth programs. For example, glucose has been shown to influence auxin distribution (via PIN2 proteins) and affect cell expansion in *Arabidopsis* root [46]. Similar results have also been reported in other species, such as in tulips, where glucose is the most advantageous mediator for shoot multiplication [47], and in *Prunus mume*, glucose also exerted better effects on direct shoot regeneration and proliferation compared with sucrose, sorbitol, and fructose [48]. These results collectively suggest that both basal salt composition and carbohydrate source are critical determinants of an efficient regeneration system for *Bombax ceiba*.

Phytohormone balance, particularly the cytokinin-to-auxin ratio, also plays a crucial role in plant regeneration [49]. Various PGRs—including 6-BA, 2,4-D, IBA, KIN, and NAA—have been evaluated in *Bombax ceiba*. Previous studies reported that 6-BA is more effective than 2,4-D in promoting callus proliferation and somatic embryogenesis, especially with continuous application at 0.44 µM 6-BA [19]. Our results corroborated these findings, showing that combinations of 0.5 mg·L^−1^ 6-BA with 0.1 mg·L^−1^ 2,4-D efficiently induced callus, while 0.5 mg·L^−1^ IBA with 1 mg·L^−1^ 6-BA promoted callus differentiation and adventitious shoots.

Our study successfully obtained regenerated plants through both direct and indirect organogenesis pathways. In the direct pathway, the differentiation efficiency reached 100%, whereas the indirect pathway only reached a differentiation rate of 51%. This discrepancy may be attributed to the lower abundance of stem cells in callus tissue. Previous research has highlighted the importance of stem cells and their regulatory networks in de novo organogenesis [15,39]. The higher regeneration efficiency in direct organogenesis likely stems from the inherently rich stem cell population and active regulatory gene expression in the shoot apical region. In contrast, callus induction requires dedifferentiation and reprogramming before shoot initiation. Therefore, future studies should focus on enhancing embryogenic callus formation by exploring diverse PGR, photoperiods, and medium components. Additionally, identifying key regulators—such as *Wuschel*-related homeobox (*WOX*), Baby Boom (*BBM*), and the AP2/ERF family of genes—may help to improve somatic reprogramming and organogenesis [50,51,52,53,54].

In vitro production of plantlets with profuse rooting is essential for the successful establishment of regenerated plants in soil. In this study, we tested a range of common rooting media and PRGs to promote rooting induction, like NAA and KIN. Previous reports on *Bombax ceiba* rooting have utilized MS supplemented with 0.5–1 mg·L^−1^ IBA. However, our findings indicate that ½ MS supplemented with 0.5 mg·L^−1^ NAA is also an effective choice to achieve a rooting rate of 96.7% within 15 days. Therefore, the rooting stage is not the critical limiting step in the micropropagation of *Bombax ceiba*.

However, plantlets produced via in vitro culture are too delicate to withstand ambient environmental conditions [55]. During the acclimatization stage, *Bombax ceiba* plantlets exhibited high sensitivity to excessive moisture, which often resulted in root rot and reduced survival. Improper control of irrigation and humidity during this stage was identified as a major factor contributing to the relatively low survival rate of 75%. Future efforts will focus on optimizing hardening-off techniques by improving drainage, regulating watering frequency, and gradually reducing ambient humidity to enhance the successful establishment of regenerated plantlets.

## 4. Materials and Methods

### 4.1. Plant Materials and Explant Preparation

*Bombax ceiba* seeds were collected in mid-April 2022 from Wu Lie Town (108.797715° E, 19.288302° N), Changjiang Li Autonomous County, Hainan Province, China. After collection, the seeds were stored dry at room temperature. Before use, the seeds were surface-sterilized under aseptic conditions by soaking in 75% ethanol for 1 min, followed by three washes with sterilized distilled water. They were then treated with 2% NaClO (Xilong Scientific, Guangdong, China) solution for 15 min and rinsed three times with sterile water, followed by immersion in 10% H_2_O_2_ (Xilong Scientific, Guangdong, China) for 1 h and final rinses with sterile water three times. The treated seeds were cultured on seedling growth medium that contained ½ MS (Murashige and Skoog 1962, 1/2 × macro and micro salt and the other parts remaining unchanged) supplemented with 2%, (*w*/*v*) glucose (Xilong Scientific, Shantou, China) and 8 g·L^−1^ carrageenan (Solarbio, Beijing, China). The growth temperature was 26–28 °C, while the relative humidity was 70–80%. The seeds were cultured in the dark for 7 days before shifting to long-day culture conditions (16 h light/8 h dark) for 10 days. The light intensity was 60 μmol⋅m^−2^·s^−1^. After 17 days of in vitro culture, the seedlings reached a height of approximately 6–8 cm and were selected for further use. The roots and cotyledons were carefully removed under sterile conditions to isolate the hypocotyl region, which was then cut into 1.0–1.2 cm segments and placed horizontally on the induction medium for callus induction. For adventitious shoot proliferation, shoot buds approximately 0.2–0.5 cm in length were excised and used as explants.

### 4.2. Medium Preparation and Tissue Culture

½ MS (coolaber, Beijing, China) medium was used for all regeneration and rooting experiments. Glucose was used as the carbohydrate source. The pH of the medium was adjusted to 5.8–6.0 using 1 M NaOH (Sinopharm Chemical Reagent Co., Ningbo, China) or HCl (Sinopharm Chemical Reagent Co., Ningbo, China) and gelled with 8 g·L^−1^ agar (Solarbio, Beijing, China) or 8 g·L^−1^ carrageenan. After autoclaving the medium at 121 °C for 20 min, plant growth regulators (PGRs) were sterilized by filtration through a 0.22 μm membrane filter and added to the medium once it cooled to approximately 50 °C. The medium was then poured into sterile tissue culture dishes or bottles, each containing 30 mL of medium.

### 4.3. Callus Induction and Differentiation

For callus induction culture, hypocotyls were transferred horizontally into callus induction medium and cultured in darkness at 26–28 °C for 4 weeks to induce callus formation. The effects of four PGRs, including 2,4-D, 6-BA, KIN, and NAA (All PGRs were obtained from Sigma-Aldrich St. Louis, MO, USA), on callus induction were tested, based on previous evidence of their benefits for callus induction in the *Malvaceae* family [17,35]. Experimental treatments included different concentrations and combinations of these PGRs (see Table 1 for details). Each treatment was replicated at least three times, with each replicate containing 30 hypocotyl explants, for a total of 90 explants. The number and frequency (%) of callus initiation were recorded after 30 days. Callus induction rate = (Number of calluses appeared on both sides of hypocotyls/Number of hypocotyls) × 100%.

To evaluate the differentiation potential of the callus, a three-factor, three-level orthogonal experiment was designed using different concentrations of IBA, KIN, and 6-BA in differentiation medium (DFM). For each treatment, 30 calluses were cultured in bottles, with 3–6 calluses per bottle, under controlled conditions of 26–28 °C, 60–70% relative humidity, and a 16 h photoperiod with a light intensity of 60 μmol·m^−2^·s^−1^. Each treatment was replicated three times. After 30 days, differentiation into bud points was recorded. The adventitious bud rate and differentiation coefficient were calculated after 45 days as follows: Callus differentiation rate = (Number of differentiated callus/Total number of callus) × 100%; Differentiation coefficient = (Number of adventitious shoots longer than 1 cm/Number of differentiated calluses) × 100%.

### 4.4. Buds Proliferation into Adventitious Shoots

Shoot buds that measured approximately 0.2–0.5 cm in length, or stem buds and lateral buds from tissue culture plantlets, were vertically inserted into suitable DFM with 3–5 buds per bottle. Each treatment was performed in at least three independent replicates, with 15 buds per replicate. The cultures were maintained at 28 °C with a relative humidity of 60–70%. A 16 h photoperiod with a light intensity of 60 μmol⋅m^−2^·s^−1^ was used for 30 days. Buds were then proliferated into adventitious shoots. The statistical proliferation rates and coefficients were determined using the following formulas: Sprouting rate = (Number of sprouting buds/Total number of buds) × 100%; Proliferation coefficient = (Number of adventitious shoots longer than 1 cm/Number of sprouting buds) × 100%.

### 4.5. Rooting of Adventitious Shoots

Adventitious shoots that showed strong growth were cut from the base and redundant calluses were removed. The cleaned shoots were then vertically inserted into rooting media, either R1 (½ MS + 0.5 mg·L^−1^ NAA + 30 g·L^−1^ glucose + 8 g·L^−1^ carrageenan, pH 5.8–6.0) or R2 (½ MS + 0.5 mg·L^−1^ NAA + 0.5 mg·L^−1^ KIN + 30 g·L^−1^ glucose + 8 g·L^−1^ carrageenan/agar, pH 5.8–6.0). Each treatment was performed in at least three independent replicates, with 15–30 adventitious shoots per replicate. Cultures were maintained under a 16 h photoperiod at 26–28 °C, with 60–70% relative humidity and a light intensity of 60 μmol·m^−2^·s^−1^. Root formation was assessed using the following formulas: Rooting rate = (Number of rooted adventitious shoots/Total Number of adventitious shoots) × 100%; Rooting coefficient = (Number of roots/Number of rooted adventitious shoots).

### 4.6. Hardening and Transfer of Regenerated Plantlets in Soil

Rooted plantlets (2–3 cm) were carefully removed from culture tubes and rinsed with tap water to remove any residual agar. The plantlets were then transplanted into plastic pots containing a substrate mixture of nutrient soil, vermiculite, and perlite (3:1:1, *v*/*v*/*v*). They were acclimated in a climate chamber at 26 °C under a 16 h light/8 h dark photoperiod, with a light intensity of 60 μmol·m^−2^·s^−1^ and relative humidity of 60–70% for 4 weeks. Subsequently, well-established plantlets were transferred to outdoor conditions for further growth.

### 4.7. Statistical Analysis

The results were recorded as the mean ± standard error (SE). GraphPad Prism 9.5 statistical software was used for statistical analysis and plotting. One-way ANOVA with multiple comparison analysis was used for significant differences, followed by Tukey’s post hoc test.

## 5. Conclusions

In summary, this study successfully established a micropropagation system for *Bombax ceiba* using a de novo organogenesis approach. Seeds were sterilized (Figure 6A) and incubated on seedling culture medium (½ MS + 20 mg·L^−1^ glucose + 8 mg·L^−1^ Carrageenan) for 7 days in the dark and then transferred to a light (16 h photoperiod) condition for an additional 10 days, which produced sterile seedlings (Figure 6B).

In the indirect organogenesis pathway, hypocotyls from sterile seedlings were cut into small segments approximately 1–1.2 cm in length and placed horizontally on CIM5 medium. Callus formation was observed after 25–30 days (Figure 6C) with an induction rate of approximately 140%. These calluses were subsequently transferred to DFM5 medium, where adventitious shoots were regenerated after approximately 45 days (Figure 6D), yielding a differentiation rate of about 51.1% (Table 3). In the direct organogenesis pathway, shoot buds (0.2–0.5 cm) were vertically cultured on DFM5 or DFM10 medium for 30–45 days to induce adventitious shoots (Figure 6E), and showed a proliferation rate of 100% and a proliferation coefficient of 3.2. The adventitious shoots were then transferred to R1 medium for rooting (Figure 6F), where regenerated plantlets (Figure 6G) were obtained after 30 days with a rooting rate of 96.67%. This entire process is summarized in Figure 6. Overall, the indirect organogenesis pathway required 85–105 days from explant to regenerated plantlets, while the direct organogenesis pathway took 45–70 days.

## Figures and Tables

**Figure 1 plants-14-02033-f001:**
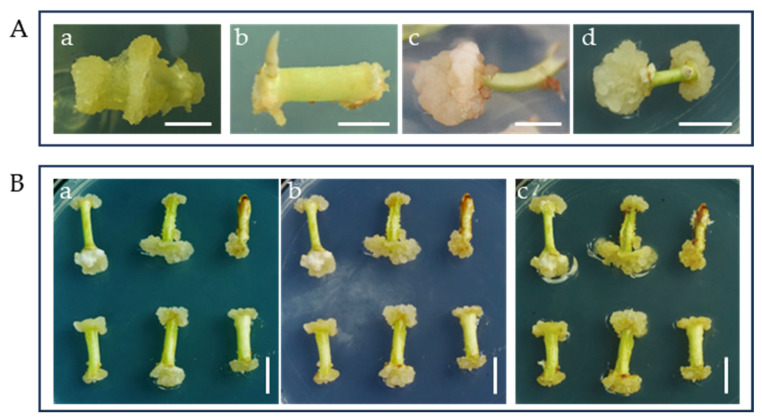
PGRs, basal medium, and treated time effects on callus induction from *Bombax ceiba* hypocotyls. (**A**) Callus induction status on different media: (**a**) CIM1, (**b**) CIM2, (**c**) CIM4, (**d**) CIM5; (**B**) Effects of CIM5 treatment time on callus status: (**a**) 25 d, (**b**) 30 d, (**c**) 35 d; Bar = 1 cm.

**Figure 2 plants-14-02033-f002:**
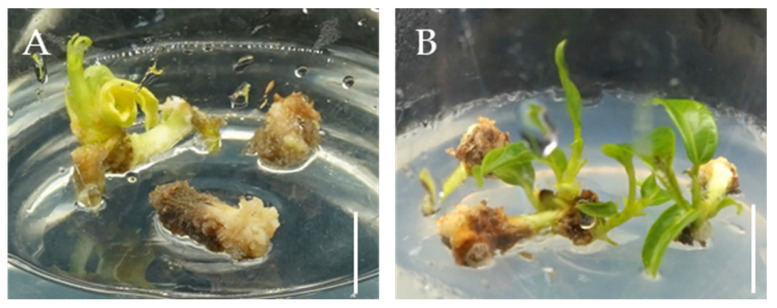
Adventitious shoots differentiated from calluses on (**A**) DFM2 and (**B**) DFM5. Bar = 1 cm.

**Figure 3 plants-14-02033-f003:**
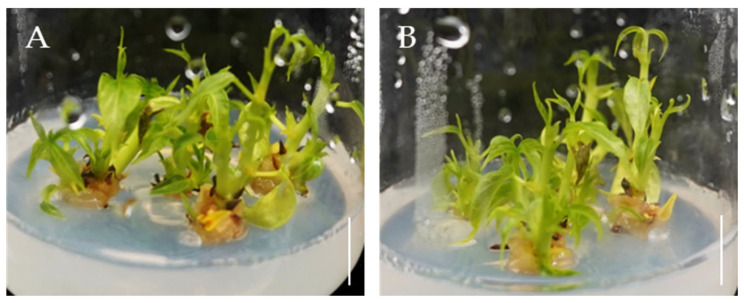
Adventitious shoots proliferated from buds after 30 days of culture on different media. Proliferation on (**A**) DFM5 and (**B**) DFM10 media; Bar = 1 cm.

**Figure 4 plants-14-02033-f004:**
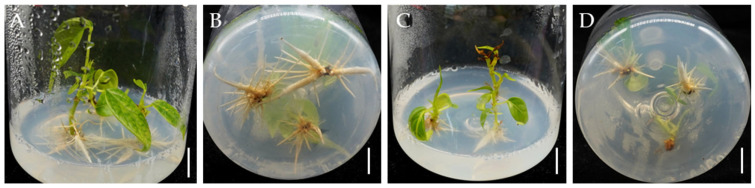
Effects of PGRs on rooting of *Bombax ceiba* adventitious shoots. (**A**) Healthy shoot growth in R1 rooting medium; (**B**) Induction of robust adventitious roots in R1 (½ MS + 0.5 mg·L^−1^ NAA) medium; (**C**) Rooting was observed in R2 medium, but shoot tip browning occurred; (**D**) Adventitious roots induced in R2 medium were short and thin; Bar = 1 cm.

**Figure 5 plants-14-02033-f005:**
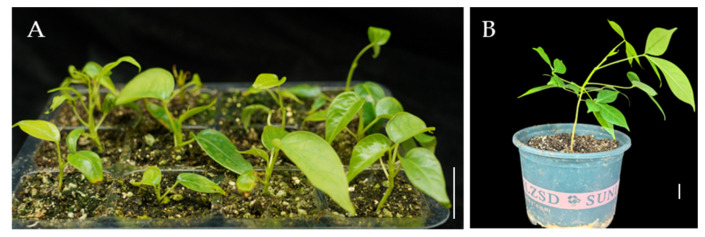
Acclimatization and transplantation of *Bombax ceiba* tissue culture-derived plantlets. (**A**) Acclimatization of plantlets in a controlled growth chamber; (**B**) Plantlets after 4 months of growth following transplantation to soil; Bar = 2 cm.

**Figure 6 plants-14-02033-f006:**
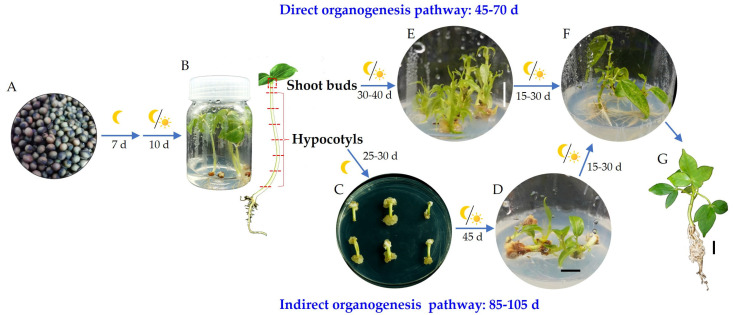
Development of the direct and indirect organogenesis-based plantlet regeneration system in *Bombax ceiba*. (**A**) Disinfection of seeds; (**B**) Sterile seedling production; (**C**) Shoot buds proliferation into adventitious shoots (direct organogenesis); (**D**) Callus induction from hypocotyl explants (indirect organogenesis); (**E**) Callus differentiation into adventitious shoots; (**F**) Rooting of adventitious shoots; (**G**) Regenerated plantlet; The moon symbol indicates dark culture condition, while the combination of moon and sun icons represents a 16 h light/8 h dark photoperiod; Bar = 1 cm.

**Table 1 plants-14-02033-t001:** Effects of basal medium and plant growth regulators (PGRs) on callus induction in *Bombax ceiba hypocotyls*.

Media	Basal Media	PGRs (mg·L^−1^)				No. of Callus Induction *	Callus Induction Rate (%) *
		2,4-D	KIN	6-BA	NAA		
CIM1	MS	0.1	0.5	0	0	18.7 ± 0.6 d	62.2 ± 3.8 d
CIM2	MS	0	0	0.5	2	0 e	0 e
CIM3	MS	0.1	0	0.2	0	26.3 ± 0.6 c	87.8 ± 3.8 c
CIM4	MS	0.1	0	0.5	0	30.7 ± 1.2 b	102.2 ± 7.7 b
CIM5	½ MS	0.1	0	0.5	0	42.0 ± 2.0 a	140.0 ± 6.7 a

The basal media MS and ½ MS refer to MS or ½ MS supplemented with 30 g·L^−1^ glucose and 8 g·L^−1^ agar. For each treatment, 30 hypocotyls were used and the callus induction rates were calculated based on both end of each hypocotyl. * Data are presented as the mean ± standard error (n = 3); different lowercase letters in each column indicate statistically significant differences; *p* < 0.001.

**Table 2 plants-14-02033-t002:** Effects of different PGRs on callus differentiation in *Bombax ceiba*.

Media	PGRs (mg·L^−1^)			Character of Callus Differentiation
	IBA	KIN	6-BA	
DFM1	0	0	0.5	Callus browning is not obvious, but no differentiation is observed.
DFM2	0	0.15	2	Slight callus browning, and differentiation is limited; only a few calluses form adventitious buds.
DFM3	0	0.5	1	Callus proliferates and browns, but no differentiation is observed.
DFM4	0.5	0	2	Callus browning is not obvious, and no differentiation occurs.
DFM5	0.5	0.15	1	Callus browns at the sprouting site and differentiates into 2–3 adventitious shoots.
DFM6	0.5	0.5	0.5	Callus turns black at the sprouting site, with no adventitious shoot formation.
DFM7	1	0	1	Callus gradually browns, and no differentiation is observed.
DFM8	1	0.15	0.5	Callus shows no proliferation and undergoes browns.
DFM9	1	0.5	2	Callus browns, with no differentiation.

**Table 3 plants-14-02033-t003:** Effects of different PGRs on callus differentiation rate and coefficient.

Media	PRGs (mg·L^−1^)			Callus Differentiation Rate (%) *	Differentiation Coefficient *
	IBA	KIN	6-BA		
DFM2	0	0.15	2	4.4 ± 1.9 b	1.3 ± 0.6 a
DFM5	0.5	0.15	1	51.1 ± 1.9 a	1.8 ± 0.1 a

* Lowercase letters in each column indicate statistically significant differences; *p* < 0.001.

**Table 4 plants-14-02033-t004:** Effect of PGRs on shoot proliferation in buds of *Bombax ceiba*.

Medium	PRGs (mg·L^−1^)			No. of Buds	Sprouting Rate (%)	No. of Proliferation Shoots ^†^	Proliferation Coefficient *
	IBA	KIN	6-BA				
DFM5	0.5	0.15	1	15	100	48	3.2 ± 0.1
				15		49	
				15		47	
DFM10	0.5	0	1	15	100	48	3.2 ± 0.0
				15		49	
				15		48	

^†^ Only adventitious shoots taller than 1 cm were counted. * No significant differences.

**Table 5 plants-14-02033-t005:** Effects of PGRs on the rooting of adventitious buds from *Bombax ceiba*.

Medium	Plant Growth Regulators	No. of Shoots	No. of Rooted Shoots	Rooting Rate (%) *	No. of Roots	Rooting Coefficient *
R1	0.5 mg·L^−1^ NAA	30	29	96.7 ± 3.3 a	79	2.7 ± 0.1 a
		30	28		78	
		15	15		40	
R2	0.5 mg·L^−1^ NAA + 0.5 mg·L^−1^ KIN	30	21	64.4 ± 5.1 b	31	1.5 ± 0.1 b
		30	19		27	
		15	9		14	

* Lowercase letters indicate statistically significant differences; *p* < 0.001.

## Data Availability

Data are contained within the article.

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
