# Peer review of "Highly Efficient Regeneration of Bombax ceiba via De Novo Organogenesis from Hypocotyl and Bud Explants"

_plants, 2025, doi:10.3390/plants14132033_

Round 1

Reviewer 1 Report (New Reviewer)

Comments and Suggestions for Authors

Dear Authors,
All my comments are marked as comments in the attached PDF of the Manuscript.

Author Response

Thank you for your thorough and thoughtful comments. We have carefully revised the manuscript according to all the suggestions.

Reviewer 2 Report (New Reviewer)

Comments and Suggestions for Authors

The article discusses the features of in vitro organogenesis of the medicinal and ornamental tree Bombax ceiba. The results showed that in vitro organogenesis de novo in B. ceiba is effective both by indirect and direct pathways. The authors identified the most effective doses of plant growth regulators and in vitro cultivation conditions that allow obtaining the largest number of new regenerated plants suitable for planting in soil. As a result, a technology for highly effective in vitro propagation of the valuable plant species B. ceiba was created.

The article contains elements of new information on in vitro clonal propagation of Bombax ceiba plants; it is written in good language, easy to read, and can be accepted for publication in the journal Plants after correcting some minor deficiencies.

There are a number of comments on the article that the authors should take into account:

  1. The Abstract (L. 25-26) states that the ‘callus induction rate of 140%’ and ‘differentiation rate of 51%’. This is causing some confusion and requires some clarification as to what exactly these percentages mean and how they were obtained. The same is probably true for L. 27-28: ‘average sprouting rate of 100%’ – if 100%, then what is the word average for? Or does the calculation method imply something else?
  2. L. 107: the text contains a repetition of the data in Table 1. The authors should rework this phrase.
  3. Table 1 contains a column for ‘No. of hypcotyls’. However, I believe that since all the values ​​in this column are the same (30), and this column could be deleted, with a heading stating that 30 explants (or hypocotyls) were taken for each basal medium. Also, the line below the table should be clarified as ‘in each column’, and a clarification should be added that the callus induction rate was calculated for both ends of each hypocotyl.
  4. L. 168-170 contains incorrect information, judging by the data in Table 4.
Comments on the Quality of English Language

No comments.

Author Response

Comments 1: The Abstract (L. 25-26) states that the ‘callus induction rate of 140%’ and ‘differentiation rate of 51%’. This is causing some confusion and requires some clarification as to what exactly these percentages mean and how they were obtained. The same is probably true for L. 27-28: ‘average sprouting rate of 100%’ – if 100%, then what is the word average for? Or does the calculation method imply something else?

Response 1: Thank you very much for your valuable feedback. we have revised the text to clarify the confusion. Specifically, to explain the unusually high callus induction rate, we added the phrase “callus was induced from both ends of each hypocotyl explant” in line 24. Additionally, the phrase “average sprouting rate of 100%” in lines 27–28 has been revised to “achieved a 100% sprouting rate” for clarity (see line 30).

Comments 2: L. 107: the text contains a repetition of the data in Table 1. The authors should rework this phrase.

Response 2: Thank you for your kind reminder. The redundancy in line 107 has been removed. please see line 97.

Comments 3: contains a column for ‘No. of hypcotyls’. However, I believe that since all the values ​​in this column are the same (30), and this column could be deleted, with a heading stating that 30 explants (or hypocotyls) were taken for each basal medium. Also, the line below the table should be clarified as ‘in each column’, and a clarification should be added that the callus induction rate was calculated for both ends of each hypocotyl.

Response 3: Thank you for your constructive suggestions. To maintain clarity and conciseness in the table heading, we have removed the column “No. of hypocotyls” and added the statement “For each treatment, 30 hypocotyls were used, and the callus induction rate was calculated based on both ends of each hypocotyl” below the table. Additionally, the statistical description has been revised to include “in each column” for improved clarity.

Comments 4: L. 168-170 contains incorrect information, judging by the data in Table 4.

Response 4: Thank you for pointing this out. The inconsistency between the text and the data in Table 4 has been corrected. The revised text can be found in lines 162–170.

Reviewer 3 Report (New Reviewer)

Comments and Suggestions for Authors

A rapid and efficient regeneration system for Bombax ceiba was developed using both indirect (hypocotyl-derived callus) and direct (shoot bud) organogenesis pathways. High callus induction (140%) and shoot regeneration rates (up to 100%) were achieved, with successful rooting and acclimatization. This system enables year-round propagation and supports future transgenic breeding efforts.

Below are the comments.

While it mentions that no rapid and efficient method has been established, the introduction fails to articulate why this is a critical bottleneck for genetic studies or large-scale applications in B. ceiba. The real urgency or impact is not well-justified.

The benefits of tissue culture, somaclonal variation, genetic transformation, and biotechnology are repeated multiple times without adding new insights. This dilutes the focus.

Too much emphasis is placed on ornamental, medicinal, and industrial uses, while the key scientific question — the development of a regeneration system — is delayed until Line 85.

Statements like "no method has yet been established" are too absolute. There are cited tissue culture reports (Lines 67–72), contradicting this. The manuscript should instead clarify what novel aspect of the regeneration system it offers (e.g., dual pathways? improved efficiency?).

While the ornamental and industrial value of B. ceiba is extensively covered, how do these uses directly relate to the current study’s objective of developing a regeneration system? Can this section be more sharply aligned with the research aim?

You claim that this regeneration system lays the foundation for molecular studies. Have you validated its genetic stability, or performed any downstream applications to support this claim?

Line 120; Why was glucose chosen over sucrose despite noting that browning differences were “not very obvious”? Can you provide data or absorbance-based browning index scores to support this switch?

Line 126-131; You state that callus initiation began at 15 days and peaked at 25 days. Did you monitor fresh/dry weight or relative growth rate to quantify this? Also, what defines “no significant proliferation” — was this visually scored or measured?

Line 165-167; You report 100% proliferation. Is this based on visual emergence of buds or quantifiable metrics like shoot length or node number? What was the sample size and replication per treatment?

Line 177; Repeated subculturing is suggested for mass propagation. Did you assess whether genetic fidelity (e.g., RAPD/SSR markers or ploidy levels) was maintained across successive subcultures? If not, how can you ensure clonal uniformity?

How you conclude without discussing your findings? Why heading conclusion is prior to discussion?

Line 266; Previous studies primarily used apical buds and immature embryos. What is the rationale for selecting hypocotyls in your work, and how might their regenerative capacity differ?

Line 275-294; How might carbohydrate metabolism influence callus induction and shoot regeneration mechanistically in B. ceiba? Your choice of ½ MS medium reduced browning compared to full-strength MS, which you attribute to salt stress. Did you measure ion concentrations or osmotic potential to support this hypothesis?

Line 328; Considering the survival rate of 75%, what are the major causes of mortality, and how could these be mitigated in future protocols?

Author Response

Comments 1:

While it mentions that no rapid and efficient method has been established, the introduction fails to articulate why this is a critical bottleneck for genetic studies or large-scale applications in B. ceiba. The real urgency or impact is not well-justified.

The benefits of tissue culture, somaclonal variation, genetic transformation, and biotechnology are repeated multiple times without adding new insights. This dilutes the focus.

Too much emphasis is placed on ornamental, medicinal, and industrial uses, while the key scientific question — the development of a regeneration system — is delayed until Line 85.

Statements like "no method has yet been established" are too absolute. There are cited tissue culture reports (Lines 67–72), contradicting this. The manuscript should instead clarify what novel aspect of the regeneration system it offers (e.g., dual pathways? improved efficiency?).

While the ornamental and industrial value of B. ceiba is extensively covered, how do these uses directly relate to the current study’s objective of developing a regeneration system? Can this section be more sharply aligned with the research aim?

You claim that this regeneration system lays the foundation for molecular studies. Have you validated its genetic stability, or performed any downstream applications to support this claim?

Responses 1: 

We thank for your thoughtful and constructive comments. In the revised Introduction, we have restructured the content to better highlight the scientific rationale and urgency for establishing an efficient regeneration system in B. ceiba. Specifically:

  1. We reduced the length of the section describing the ornamental and industrial uses, while slightly expanding the sentence introducing the species' ornamental value in response to a reviewer’s request. Meanwhile, we placed greater emphasis on the significance of regeneration systems for genetic transformation and breeding applications.
  2. We clarified that while prior tissue culture work has been reported, most studies were limited to shoot proliferation using shoot tips or immature embryos, without establishing a comprehensive system that integrates both indirect and direct regeneration pathways.
  3. The novelty of our work lies in the use of two explant types (hypocotyls and shoot buds) and the development of a dual-pathway system that is efficient, reproducible, and suitable for downstream genetic applications.
  4. We also acknowledged the limitation that genetic stability or transgenic validation was not performed in this study but emphasized that the regeneration system provides a reliable technical platform for such future applications.

These changes address the concerns raised and align the Introduction more closely with the objectives and significance of our study.

comment 2: Line 120; Why was glucose chosen over sucrose despite noting that browning differences were “not very obvious”? Can you provide data or absorbance-based browning index scores to support this switch?

Response 2: Thank you for your insightful comment. Although we did not quantify the degree of browning using a browning index or absorbance-based measurements in this particular experiment, our decision to use glucose over sucrose as the carbohydrate source was based on consistent trends observed in our previous seedling growth and preliminary callus induction tests. In these experiments, explants cultured on glucose-containing media exhibited better overall tissue viability and growth, with a lower tendency to brown, compared to those cultured on sucrose-containing media. While the visual difference in browning during the callus induction stage was not striking, the use of glucose contributed to improved culture performance and reduced phenolic oxidation in practice.

We agree that quantitative data would provide stronger support for this observation and will consider incorporating such measurements in future studies.

comment 3: Line 126-131; You state that callus initiation began at 15 days and peaked at 25 days. Did you monitor fresh/dry weight or relative growth rate to quantify this? Also, what defines “no significant proliferation” — was this visually scored or measured?

Response 3: Thank you for your valuable comment. In this study, callus proliferation was assessed based on consistent visual observations across three independent replicates. We defined "no significant proliferation" as the stage at which there was no apparent increase in callus size or density compared to the previous time points. While we did not quantify callus growth using fresh or dry weight measurements or calculate relative growth rates, the changes in callus initiation and proliferation were clearly distinguishable from photographic records and repeated observations.

We acknowledge the importance of quantitative measurements for more precise assessment and will consider incorporating such analyses in future studies.

Comment 4: Line 165-167; You report 100% proliferation. Is this based on visual emergence of buds or quantifiable metrics like shoot length or node number? What was the sample size and replication per treatment?

Response 4: Thank you for your thoughtful and rigorous comment. In our system, both the adventitious shoots derived from callus and those from shoot buds typically develop as independent, well-separated structures. As can be seen in the representative images, these shoots are morphologically distinct, and no tiny or undeveloped buds were observed at the time of evaluation. Following your suggestion, we would like to clarify that the reported 100% proliferation rate was based on the visual emergence of clearly distinguishable shoots with a height greater than 1 cm. This criterion has now been explicitly stated in the Materials and Methods section.

The proliferation rate was calculated based on three biological replicates, with 15 explants per treatment. We appreciate your comment and agree that including this level of detail enhances the clarity and reproducibility of our findings.

Comment 5: Line 177; Repeated subculturing is suggested for mass propagation. Did you assess whether genetic fidelity (e.g., RAPD/SSR markers or ploidy levels) was maintained across successive subcultures? If not, how can you ensure clonal uniformity?

Response 5: Thank you for your valuable comment. We did not assess genetic fidelity using molecular markers (such as RAPD/SSR) or flow cytometry in this study. However, in plant biotechnology, minor somaclonal variations can occasionally arise during repeated subculturing. As long as these variants display desirable phenotypic traits, they may still be acceptable—particularly for ornamental or medicinal applications, where slight variations in morphology or secondary metabolite profiles do not necessarily compromise their value.

In the case of Bombax ceiba, both flower traits and bioactive compound production are important application targets. Therefore, maintaining strict genetic uniformity may not always be essential at the early propagation stage, especially when phenotypic screening can be applied later. Nonetheless, we fully acknowledge the importance of clonal fidelity for large-scale propagation and commercial use. Future work will incorporate molecular or cytological methods to assess genetic stability during subculture.

Comment 6: How you conclude without discussing your findings? Why heading conclusion is prior to discussion?

Response 6: Thank you for your comment. In the original version, the Conclusion section was placed before the Discussion to provide a concise summary of the key outcomes and the practical value of the proposed method. We believed this structure might help readers quickly grasp the overall findings and their applications.

However, we fully agree that, according to conventional scientific writing standards, the Discussion should precede the Conclusion. We have now revised the manuscript accordingly to follow the standard structure.

Comment 7: Line 266; Previous studies primarily used apical buds and immature embryos. What is the rationale for selecting hypocotyls in your work, and how might their regenerative capacity differ?

Response 7: Thank you for your insightful comment. The selection of hypocotyls is closely related to our long-term goal of establishing a genetic transformation system for B. ceiba, although transformation itself is beyond the scope of this particular study. Hypocotyls contain a high proportion of actively dividing cells, making them an excellent explant source for in vitro regeneration and large-scale propagation.

Moreover, hypocotyl-based regeneration and transgene systems have been successfully applied in many other plant species, such as tomato, rapeseed, and cotton. We have added this rationale and supporting references in the revised manuscript (see lines 238–241 of the Discussion section).

Comment 8: Line 275-294; How might carbohydrate metabolism influence callus induction and shoot regeneration mechanistically in B. ceiba? Your choice of ½ MS medium reduced browning compared to full-strength MS, which you attribute to salt stress. Did you measure ion concentrations or osmotic potential to support this hypothesis?

Response 8: Thank you for this insightful question. We agree that both osmotic stress and ionic strength likely play roles in the browning and regenerative responses observed in B. ceiba. The reduced browning and improved callus induction on ½ MS medium suggest that full-strength MS may impose higher osmotic or ionic stress on the explants, leading to tissue damage and oxidation.

Although we did not directly measure ion concentrations or osmotic potential in this study, the observed phenotypes are consistent with previous reports where reduced salt strength alleviated stress-induced browning in tissue cultures of other species. We have revised the manuscript to clarify this interpretation and acknowledge the need for future physiological and biochemical analyses to confirm the mechanisms involved.

Comment 9: Line 328; Considering the survival rate of 75%, what are the major causes of mortality, and how could these be mitigated in future protocols?

Response 9: Thank you for your comment. The 75% survival rate during acclimatization was mainly affected by water management. B. ceiba plantlets showed high sensitivity to excess moisture, often resulting in root rot and tissue collapse. Inadequate control of humidity and irrigation contributed to plantlet loss.

Future work will focus on improving hardening conditions through better drainage, controlled watering, and gradual humidity reduction. A standardized acclimatization protocol will help increase the survival of transplanted plantlets.

Finally, we sincerely thank you for providing us with so many valuable comments.

Round 2

Reviewer 3 Report (New Reviewer)

Comments and Suggestions for Authors

The author has revised the MS as per the suggestions.

This manuscript is a resubmission of an earlier submission. The following is a list of the peer review reports and author responses from that submission.

Round 1

Reviewer 1 Report

Comments and Suggestions for Authors

The in vitro propagation of woody plants is worthy of investigation, even though there are some previous reports on this species. The authors followed a typical step-by-step approach in micro-propagation.  

In Lines 179-197, the authors argue that they developed two pathways: somatic embryogenesis and organogenesis. However, I am not sure about that issue. I cannot see any evidence of somatic embryogenesis.  If the callus directly developed into shoots, it would be (typical) "organogenesis" (Figure 3A-d).

The other pathway, organogenesis (termed by the authors in Line 190), seems  "direct organogenesis" or adventitious shoot formation.  (In fact, the title says somatic embryogenesis and direct organogenesis. The authors overlapped organogenesis and direct organogenesis.) 

In detail, the title of section 2.2 is recommended to be changed.

The photography of callus differentiation (Tables 2 and 3) and organogenesis (Table 4) are needed.

Figure 3 A and B can be merged as a figure.

The readers may expect a more intensive Discussion on the mechanism of the events. 

Comments on the Quality of English Language

English is good for following the manuscript.

Author Response

Comments 1: In Lines 179-197, the authors argue that they developed two pathways: somatic embryogenesis and organogenesis. However, I am not sure about that issue. I cannot see any evidence of somatic embryogenesis.  If the callus directly developed into shoots, it would be (typical) "organogenesis" (Figure 3A-d).

Response 1: Thank you very much for your insightful comment. Upon reviewing the relevant literature and re-evaluating our results, we found no clear morphological or histological evidence supporting somatic embryogenesis or the formation of distinct somatic embryos in our system. Instead, the adventitious shoots originated indirectly from callus tissue, which itself developed from hypocotyl explants. Therefore, the regeneration pathway should be more accurately described as indirect organogenesis. We have revised the terminology throughout the manuscript to reflect this correction.

Comments 2: The other pathway, organogenesis (termed by the authors in Line 190), seems "direct organogenesis" or adventitious shoot formation. (In fact, the title says somatic embryogenesis and direct organogenesis. The authors overlapped organogenesis and direct organogenesis.)

Response 2: Thank you for your valuable comments. We have defined "organogenesis" throughout the manuscript as "direct organogenesis" and have revised the title accordingly to reflect this clarification. Additionally, to improve clarity and consistency, we have replaced the term "somatic embryogenesis pathway" with "indirect organogenesis pathway" throughout the manuscript.

Comments 3: In detail, the title of section 2.2 is recommended to be changed.

Response 3: Agree. The title has been changed to “The effect of PGRs on callus differentiation”

Comments 4: The photography of callus differentiation (Tables 2 and 3) and organogenesis (Table 4) are needed.

Response 4: Thank you for your valuable suggestion. We acknowledge this oversight in our experiments. To clarify, the results presented in Table 2 come from a random testing experiment designed to test which PGRs could induce callus differentiation into adventitious shoots. As such, we documented only the experimental outcomes and did not retain photographic evidence. Figures 2 and 3 correspond to the images illustrating the results shown in Tables 3 and 4, respectively.

Comments 5: Figure 3 A and B can be merged as a figure.

Response 5: Thanks, it has been merged.

Comments 6: The readers may expect a more intensive Discussion on the mechanism of the events. 

Response 6: The discussion has been expanded accordingly. Please refer to the revised Discussion section for the additional content.

Reviewer 2 Report

Comments and Suggestions for Authors

Dear authors.

You are presenting a huge work done by you in order develop a protocol for rapid propagation of B.ceiba.  Along the text, I commented on specific points. Please relate to each of them.

Here are some general comments:

The English in the written text must be re-edited. In some cases wrong words/terms were used.

In many cases along M&M (especially) and Resutls sections, data are repeated. These sections should be rearranged to avoid it.  

Please note the comments on the titles of sections and tables.

I recommend  a major revision before accepting the article for publication.

Comments on the Quality of English Language

English editing is required.

Author Response

Comments 1: The English in the written text must be re-edited. In some cases wrong words/terms were used.

In many cases along M&M (especially) and Resutls sections, data are repeated. These sections should be rearranged to avoid it.  

Please note the comments on the titles of sections and tables.

I recommend a major revision before accepting the article for publication.

Response 1: Thank you very much for your valuable suggestions and detailed comments. We have thoroughly revised the entire manuscript and addressed all the issues you pointed out. Revisions have been highlighted in blue for your convenience.

Reviewer 3 Report

Comments and Suggestions for Authors

The paper describes two systems of micropropagation developed for Bombax ceiba, a tropical species used for multiple purposes. The authors presented interesting and novel results that contribute to the technology of this species’ propagation, although acclimatization of in vitro plants under ex vitro conditions was not tested. The study fits into the scope of the journal and may be published after minor revision as below.

Major comments:

From the Materials and Methods, it is not clear how many replications were made in each experiment and how many explants were used per each treatment conditions. Please specify.

Table 1. It is not clear from the table or from Materials and Methods what is “No. of callus induction”. Please specify.

The subheading 2.2 might be generalized as, for example, “The effect of growth regulators on callus differentiation” without specifying the concentrations of growth regulators.

In Table 5, “Rooting number” but in Materials and Methods the authors described “rooting coefficient”. If this is the same parameter, please unify between the Table and the method description.

Minor comments:

Figure 2. Please describe R1 medium in the legend.

Sometimes in the text the authors use “growth regulators” and sometimes “hormones”. It is better to choose one term and use it continuously throughout the text.

There are some typos and misusing of words. For example, Line 56 – did the authors mean “advantageous” instead of “adventitious”?

The last paragraph of the Introduction describes the results of the paper and is better to move to the end as the Conclusion.  Introduction should be finished with a brief statement of the research goals of the study.

In the Abstract, instead of the sentence “Our novel regeneration system maximizes the use 23 of sterile seedling explants.”, please give the number of rooted plants in vitro that could be obtained from each type of the explants using the developed methods.

Comments on the Quality of English Language

The quality of the language did not limit my understanding of the results although there are some typos and misusing of words (see the comments)

Author Response

Comments 1: From the Materials and Methods, it is not clear how many replications were made in each experiment and how many explants were used per each treatment conditions. Please specify.

Response 1: Thank you for your comment. The information has been added in the Materials & Methods section. See lines 363 and 370 with blue text highlighted in yellow.

Comments 2: Table 1. It is not clear from the table or from Materials and Methods what is “No. of callus induction”. Please specify.

Response 2: This has been clarified. Please see line 366, which is highlighted in yellow in the Materials & Methods section.

Comments 3: The subheading 2.2 might be generalized as, for example, “The effect of growth regulators on callus differentiation” without specifying the concentrations of growth regulators.

Response 3: Thank you for the correction. It has been revised accordingly.

Comments 4: In Table 5, “Rooting number” but in Materials and Methods the authors described “rooting coefficient”. If this is the same parameter, please unify between the Table and the method description.

Response 4: Revised as requested; please see section 5.5 with blue text highlighted in yellow.

Minor comments:

Comments 5: Figure 2. Please describe R1 medium in the legend.

Response 5: Already described; see line 199.

Comments 6: Sometimes in the text the authors use “growth regulators” and sometimes “hormones”. It is better to choose one term and use it continuously throughout the text.

Response 6: All instances of "hormones" have been replaced with "PGRs" throughout the manuscript.

Comments 7: There are some typos and misusing of words. For example, Line 56 – did the authors mean “advantageous” instead of “adventitious”?

Response 7: Thanks, revised as requested; see line 70.

Comments 8: The last paragraph of the Introduction describes the results of the paper and is better to move to the end as the Conclusion.  Introduction should be finished with a brief statement of the research goals of the study.

Response 8: Revised as requested

Comments 9: In the Abstract, instead of the sentence “Our novel regeneration system maximizes the use 23 of sterile seedling explants.”, please give the number of rooted plants in vitro that could be obtained from each type of the explants using the developed methods.

Response 9: It is difficult to provide an exact number of rooted plantlets, as both shoot buds and hypocotyl-derived adventitious shoots root well in vitro. Each shoot bud produces about 3 adventitious shoots (Proliferation coefficient: 3.2). For hypocotyls, one seedling yields 5–8 segments, with a 140% callus induction rate, 51% differentiation rate, and a differentiation coefficient of 1.8, resulting in approximately 6–10 adventitious shoots. Since these shoot buds can be further proliferated, the total number of rooted plantlets cannot be accurately determined.